biochemistry/plant science

cyanobacteria, photosynthesis, thylakoid membrane, supercomplex, mass spectrometry, atomic force microscopy

**Author for correspondence:**
Lu-Ning Liu
e-mail: luning.liu@liverpool.ac.uk

# Characterizing the supercomplex association of photosynthetic complexes in cyanobacteria

## Zimeng Zhang[1], Long-Sheng Zhao[1,2] and Lu-Ning Liu[1,3]

[1]Institute of Systems, Molecular and Integrative Biology, University of Liverpool, Crown Street, Liverpool L69 7ZB, UK
[2]State Key Laboratory of Microbial Technology, and Marine Biotechnology Research Center, Shandong University, Qingdao 266237, People's Republic of China
[3]College of Marine Life Sciences, and Frontiers Science Center for Deep Ocean Multispheres and Earth System, Ocean University of China, Qingdao 266003, People's Republic of China

L-NL, 0000-0002-8884-4819

The light reactions of photosynthesis occur in thylakoid membranes that are densely packed with a series of photosynthetic complexes. The lateral organization and close association of photosynthetic complexes in native thylakoid membranes are vital for efficient light harvesting and energy transduction. Recently, analysis of the interconnections between photosynthetic complexes to form supercomplexes has garnered great interest. In this work, we report a method integrating immunoprecipitation, mass spectrometry and atomic force microscopy to identify the inter-complex associations of photosynthetic complexes in thylakoid membranes from the cyanobacterium *Synechococcus elongatus* PCC 7942. We characterize the preferable associations between individual photosynthetic complexes and binding proteins involved in the complex–complex interfaces, permitting us to propose the structural models of photosynthetic complex associations that promote the formation of photosynthetic supercomplexes. We also identified other potential binding proteins with the photosynthetic complexes, suggesting the highly connecting networks associated with thylakoid membranes. This study provides mechanistic insight into the physical interconnections of photosynthetic complexes and potential partners, which are crucial for efficient energy transfer and physiological acclimatization of the photosynthetic apparatus. Advanced knowledge of the protein organization and interplay of the photosynthetic machinery will inform rational design and engineering of artificial photosynthetic systems to supercharge energy production.

# 1. Introduction

Photosynthesis is the natural process conducted by plants, algae and cyanobacteria to transform light energy into chemical energy to drive the biochemistry of life on the Earth. Photosynthesis is performed in two steps: the light-dependent reactions and the light-independent reactions. In most cyanobacteria, light-dependent reactions of photosynthesis occur in the specialized intracellular membranes termed thylakoid membranes. Unlike the thylakoid membranes of higher plants, cyanobacterial thylakoids are not differentiated into grana and stroma regions; instead, they generally form stacks of membrane layers that sit between the cytoplasmic membrane and central cytoplasm [1,2].

A unique feature of the cyanobacterial thylakoid membrane is that it provides a membrane platform accommodating the protein complexes of both photosynthetic and respiratory electron transfer chains [2,3]. The physiological functions and coordination of electron transport components are fundamental for efficient electron flow and bioenergetic modulation, enabling cyanobacterial cells to thrive in distinct ecological niches. Over the past decades, the atomic structures of major photosynthetic membrane complexes from cyanobacteria have been resolved, including Photosystem I (PSI) [4–6], Photosystem II (PSII) [7–10], Cytochrome $b_6f$ (Cyt $b_6f$) [11] and photosynthetic Complex I – NAD(P)H dehydrogenase (NDH-1) [12–16]. These studies have provided structural insight into electron transport mechanisms within individual bioenergetic complexes. Moreover, increasing experimental evidence has been achieved to demonstrate the close associations of photosynthetic complexes in cyanobacteria, leading to the occurrence of photosynthetic supercomplexes to promote inter-complex electron transport. These supercomplexes involve the PSII–PSI supercomplex [17], the PSII–PSI–phycobilisome megacomplexes [18] and the PSI–NDH-1 supercomplex [19]. Our recent atomic force microscopy (AFM) observations have revealed explicitly the strong lateral associations of different photosynthetic complexes in cyanobacterial thylakoid membranes [20]. Likewise, photosynthetic supercomplexes have also been found in the chloroplasts of green algae and plants [21,22]. The structural and functional associations of different photosynthetic complexes in the thylakoid membrane are crucial for efficient electron transport between complexes and in the overall electron flow pathways.

Despite the substantial studies, our understanding of how photosynthetic complexes interact and cooperate with others to fulfil efficient electron transport is still rudimentary. The inherent challenges are the organizational heterogeneity and dynamics of electron transport chains in cyanobacterial thylakoid membranes, suggesting the transient and flexible interactions between different photosynthetic membrane complexes [2,20,23]. Here, we describe an approach that combines immunoprecipitation, mass spectrometry and AFM to identify the interactions between photosynthetic complexes in thylakoid membranes from a model cyanobacterium *Synechococcus elongatus* PCC 7942 (Syn7942) and the specific binding sites involved in inter-complex associations. The results provide insight into the structural interconnections of electron transport complexes in thylakoid membranes, which are pivotal for photosynthetic electron flow and regulation.

# 2. Material and methods

## 2.1. Strains and cell culture

The wild-type (WT) cyanobacterial strain *Synechococcus elongatus* PCC 7942 (Syn7942) was used in this study to generate the enhanced green fluorescence protein (GFP)-labelled strains. The GFP-labelled Syn7942 strains have been generated in our previous study [3]. In brief, the fluorescently labelled strains were generated by inserting GFP and apramycin region amplified from the plasmid pIJ786 to the 3′ end of *psaE* of PSI, *psbB* of PSII, *petA* of Cyt $b_6f$ and *atpB* of ATPase [3,24–27], using the Redirect strategy [28,29]. All cell cultures were incubated in BG11 medium at 30°C under $40 \, \mu\text{E} \, \text{m}^{-2} \, \text{s}^{-1}$ white illumination in Nunc™ Cell Culture Treated TripleFlasks with constant shaking. Supplementary $50 \, \mu\text{g} \, \text{ml}^{-1}$ of antibiotics were added to the medium for mutated strains.

## 2.2. Thylakoid membrane preparation and solubilization

WT and GFP-labelled Syn7942 cells were harvested when OD750 reaches between 0.6 and 1.0. Syn7942 cells were pelleted by centrifugation at 4000 g for 10 min at 4°C and washed with buffer A (50 mM MES-NaOH, pH 6.5, 5 mM $CaCl_2$, and 10 mM $MgCl_2$). Cell pellets were resuspended in

buffer A containing 25% glycerol and were broken by glass bead (212–300 µm in diameter, Sigma-Aldrich) at 4°C with vortex at 2700 r.p.m. for five times 1 min on and 1 min off and then 10 times 30 s on and 30 s off. Crude thylakoid membrane fractions were prepared by centrifugation [30] and were resuspended in membrane resuspension buffer (10 mM Tris pH 6.8, 200 mM NaCl, 1 mM EDTA) to obtain a 200 µg ml$^{-1}$ chlorophyll (Chl) concentration. The thylakoid membrane fractions were then solubilized by digitonin (Sigma-Aldrich) at various concentrations (0–2%) for 30 min at room temperature and were shaken on a vortex at 600 r.p.m. The solubilized thylakoid membrane proteins were purified by centrifugation at 40 000 g for 30 min followed by centrifugation at 21 100 g for 20 min. To ensure both membrane solubilization and intactness of membrane complexes, the supernatant samples resulting from the treatment of different digitonin concentrations were examined by negative-staining transmission electron microscopy (data not shown). Subsequently, 1% digitonin was applied to solubilize thylakoid membranes and the supernatant was then applied for GFP pull-down assays.

## 2.3. Green fluorescence protein pull-down assays

GFP pull-down assays were carried out using µMACS and MultiMACS GFP Isolation Kits (Miltenyi Biotec). Thylakoid membrane protein samples with 10 µg Chl $a$ were incubated with 50 µl beads and membrane resuspension buffer was added to reach a total volume of 200 µl. The columns were prewashed with lysis buffer from the kit. After loading samples, the columns were further washed five times with membrane resuspension buffer mentioned above before elution. For SDS-PAGE analysis, proteins were eluted with 50 µl elution buffer following the instructions provided by the manufacturer, and 10 µl of samples were loaded onto a 12% SDS gel with 4 × sample buffer (1.57% Tris–HCl pH 6.8, 4% SDS (w/v), 20% glycerol, 0.1% bromophenol blue (w/v), 1.5% dithiothreitol (w/v)). For mass spectrometry, bound proteins and beads were collected by taking the column out of the magnetic field and were washed with 50 µl membrane resuspension buffer. Ten microlitres was used for SDS-PAGE and the remaining 40 µl was used for proteomic analysis.

## 2.4. Mass spectrometry

Bound proteins and beads were resuspended in 50 µl of AMBIC containing 0.05% (w/v) Rapigest and heated to 80°C for 10 min to facilitate the elution of GFP-labelled proteins and interactants. Cysteine reduction was performed by incubating with 5 µl of a 9.2 mg ml$^{-1}$ DTT solution and at 60°C for 10 min. Subsequent alkylation was carried out by dark incubation with 5 µl of a 33 µg ml$^{-1}$ iodoacetamide for 30 min. In-solution digestion was carried out by incubating with 200 ng of trypsin at 37°C overnight. The reaction was quenched by the addition of 0.5 µl of trifluoroacetic acid (TFA) at 37°C for 45 min. RapidGest degradation products and any insoluble material were removed by centrifugation at 16 000 g for 20 min at 4°C. Injected samples (10 µl for each) were analysed using an Ultimate™ 3000 RSLCnano system (Thermo Fisher Scientific, Hemel Hempstead, UK) coupled with a Q Exactive™ HF mass spectrometer (Thermo Fisher Scientific, Hemel Hempstead, UK). The samples were loaded onto a trapping column (Thermo Fisher Scientific, PepMap™ 100, C18, 300 µm × 5 mm) using partial loop injection for 7 min at a flow rate of 4 µl min$^{-1}$ with 0.1% (v/v) formic acid (FA). The samples were resolved on the analytical column (EASY-Spray™ C18, 75 µm × 500 mm, particle size 2 µm column) using a gradient of 97% A (0.1% FAFA), 3% B (99.9% ACN, 0.1% FA) to 60% A, 40% B over 30 min at a flow rate of 300 nl min$^{-1}$. The data-dependent program used for data acquisition consisted of a 60 000 resolution full-scan MS scan (AGC set to 3e$^6$ ions with a maximum fill time of 100 ms), the 10 most abundant peaks were selected for MS/MS using a 30 000 resolution scan (AGC set to 1e$^4$ ions with a maximum fill time of 45 ms) with an ion selection window of 1.2 m z$^{-1}$ and normalized collision energy of 30. To avoid repeated selection of peptides for MS/MS the program used a 20 s dynamic exclusion window.

### 2.4.1. Progenesis data analysis

The raw liquid chromatography–mass spectrometry (LC-MS) files were analysed in Progenesis QI for Proteomics label-free analysis software, which aligned the files and peak picks for quantification by peptide abundance. For the comparison, five groups were created (all groups; WT versus PSI; WT versus PSII; WT versus Cyt $b_6f$; WT versus ATPase) containing the respective replicate samples. The software first aligned the LC-MS files and peak picks the aligned peptides. An aggregate file was

generated that contains all the peaks from all runs in an experiment so that there are no missing values. At this stage in the process, normalization was performed using the 'normalize against all proteins' option. The software assumed that most proteins are not changing in abundance and normalization factors are used to adjust peptide intensities.

The peptide list was exported into Peaks and MASCOT and was searched against the *Synechococcus elongatus* PCC 7942 Uniprot database (2657 proteins), and was then manually searched against a small database containing the sequence of all subunits of photosynthetic complexes (with carbamidomethyl cysteine as a fixed modification and methionine oxidation as a variable modification), and the peptide lists were imported back into Progenesis and assigned to features. Peptide matches in *peaks* were set to a 1% false discovery rate (FDR) and peptides are then filtered in Progenesis at the peptide cut-off threshold score for a 1% FDR.

## 2.5. Mass spectrometry data analysis

Ratios of complexes involved in supercomplex formation were calculated by comparing the abundance of specific protein subunits between the GFP-labelled strain and other strains in which the specific subunits were not labelled with GFP using GFP-normalized data. Since the GFP abundance of these two groups has already been normalized, theoretically, the copy number of PSI in the PSI–GFP sample is approximately the same as the copy number of PSII in the PSII–GFP samples. If the 100% supercomplexes and 1:1 ratio hypotheses were valid, it would be approximately equal to the copy number of PSI in the PSII–GFP sample. For example, if 100% PSII formed stable supercomplexes with PSI with a ratio of 1:1, the abundance of PSI in the GFP–PSII sample would be similar to the abundance of PSI in the PSI–GFP sample. As mentioned in the Results and discussion section, the quantities of PSI, PSII, Cyt $b_6f$ and ATPase were presented by the sums of the scores of all observed protein subunits. Then, the sums of value were used to carry on the subsequent ratio calculation. For example, the ratio of PSII that forms supercomplex with PSI was calculated by the sum of PSI subunits' abundance in the PSII–GFP sample divided by the sum of PSI subunits' abundance in the PSI–GFP group. Student's *t*-test was deployed to determining if there is a significant difference between the means of the two samples.

To probe the interacting protein subunits that associate with the GFP-labelled complex, the relative abundance of each subunit was calculated by determining the ratio of the abundance score of the same subunit in a specific sample to that in the sample in which GFP was labelled to the photosynthetic complex that this subunit belongs to, e.g. PsaA in the PSII–GFP pull-down sample was divided by PsaA in the PSI–GFP sample. Subsequently, statistical analysis was performed using Student's *t*-test to compare the ratio of each subunit with the ratios of other subunits within the same photosynthetic complexes in the same pull-down sample. For example, the ratio of PsaA was compared with those of PsaB, PsaC, PsaD, PsaE, PsaF, PsaJ, PsaK and PsaL as a whole.

## 2.6. Atomic force microscopy

For AFM imaging on pull-down samples, 2 µl of protein samples were adsorbed onto freshly cleaved mica surface with 38 µl of adsorption buffer (10 mM Tris–HCl pH 7.5, 150 mM, KCl, 25 mM MgCl$_2$) at room temperature for 1 h. After adsorption, the sample was carefully rinsed with 800 µl imaging buffer (10 mM Tris–HCl pH 7.5, 150 mM KCl). These buffers could ensure the electrostatically balanced interactions between AFM tip and biological samples and thereby high-resolution AFM topographs in aqueous solution [3,31–35]. AFM imaging was performed in ScanAsyst Air mode in air at room temperature using a Multimode Nanoscope VIII AFM (Bruker) equipped with a J-scanner and ScanAsyst-Air-HR (0.4 N m$^{-1}$, Bruker) at a scan frequency of 1 Hz using optimized feedback parameters and a resolution of 512 × 512 pixels. Images were processed with NanoScope Analysis software (Bruker).

For AFM imaging on native thylakoid membranes, crude thylakoid membrane fractions were further purified using a step sucrose gradient (2.0 M, 1.3 M, 1.0 M, 0.5 M) in 50 mM MES-NaOH pH 6.5, 5 mM CaCl$_2$ and 10 mM MgCl$_2$, and were centrifuged at 36 200 r.p.m. in Beckman RPS40 rotor for 1 h at 4°C, as reported previously [3,20]. The Chl-enriched samples were collected and characterized by high-resolution AFM imaging in liquid at room temperature in AC imaging mode using a NanoWizard 3 AFM (JPK) using Ultra-Short Cantilever probes (0.3 MHz, 0.3 N m$^{-1}$, NanoWorld) with optimized feedback parameters and a resolution of 512 × 512 pixels [20]. Images were processed with JPK SPM Data Processing (JPK) and ImageJ [36]. No detergent was added during membrane isolation and AFM imaging to ensure the physiological organization of isolated thylakoid membranes [37].

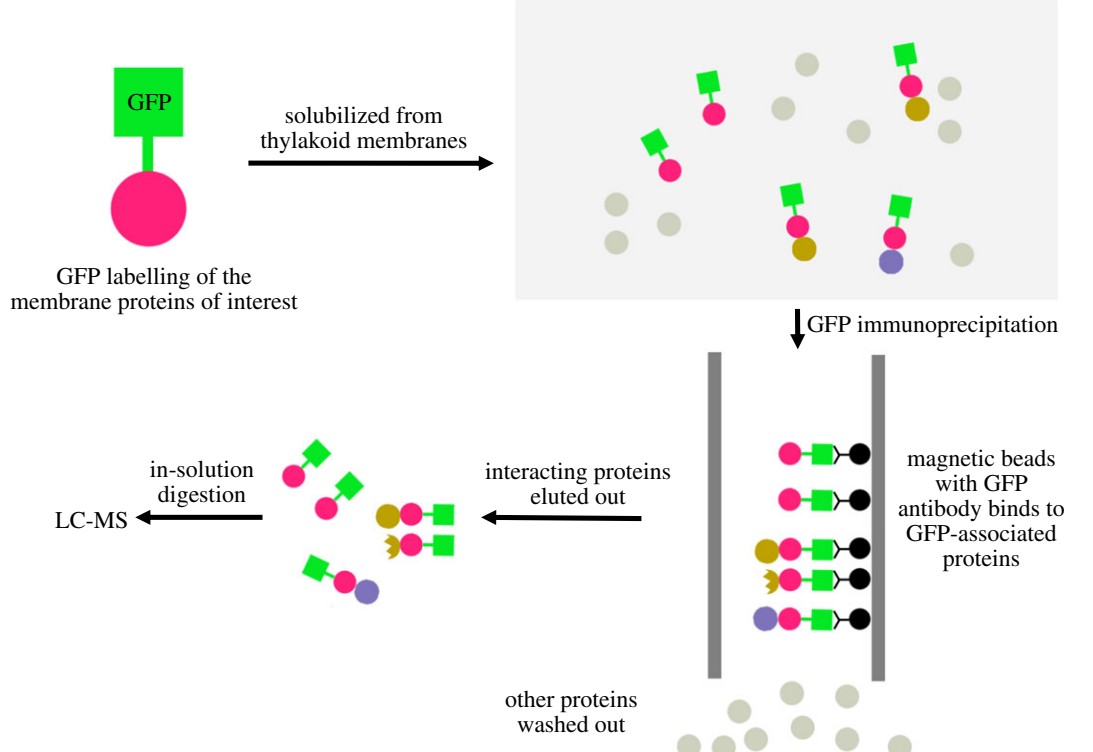

**Figure 1.** Overview of GFP immunoprecipitation methodology. Crude thylakoid membranes from GFP-labelled PSI, PSII, Cyt $b_6f$, ATPase and WT strains were prepared by centrifugation and were solubilized by 1% digitonin. Proteins that interact with GFP-labelled proteins are immunoprecipitated with GFP pull-down assay and subjected to mass spectrometric analysis. Triplicated experiments were conducted for each strain.

# 3. Results and discussion

## 3.1. Pull-down assays to study the associations of photosynthetic complexes

In the previous study, we have tagged PSI, PSII, Cyt $b_6f$ and ATPase individually with enhanced green fluorescent protein (GFP) to visualize the localization of photosynthetic complexes in Syn7942 [3]. The subunits labelled with GFP were PsaE for PSI, PsbB (CP47) for PSII, PetA for Cyt $b_6f$ and AtpB ($\beta$) for ATPase. To study the inter-complex assembly of photosynthetic complexes in thylakoid membranes, we isolated the thylakoid membranes from these Syn7942 strains and solubilized thylakoid membranes using digitonin at the optimized 1% (see Material and methods). The membrane-bound protein complexes tagged with GFP and their associated proteins/complexes were purified using GFP pull-down assays [38–40] (figure 1).

The resulting protein mixtures were characterized by AFM. AFM revealed that the protein complexes appeared as individual complexes or small assemblies, instead of forming large aggregates and membrane fragments (figure 2*a,b*). PSI trimers can be readily recognized by AFM, with a height of $5.7 \pm 0.5$ nm ($n = 21$) and a distance of $9.6 \pm 0.8$ nm ($n = 21$) between its two vertexes (figure 2*c,e*), consistent with previous studies [3,20]. The dimeric structures, putatively PSII dimers, possess a height of $5.6 \pm 0.5$ nm ($n = 11$) and a vertex distance of $10.4 \pm 1.16$ nm ($n = 11$) between two protrusions (figure 2*d,e*). Complex assemblies were often discerned by AFM (figure 2*a,b*, white circles), providing the opportunity for us to explore the interactions of different photosynthetic supercomplexes [18,41].

SDS-PAGE revealed that the GFP-labelled complexes were unambiguously present at a great abundance in each pull-down sample (figure 3). With the fusion of enhanced GFP, the PsaE, PsbB, PetA and AtpB bands were up-shifted. It was shown that the PSI and PSII pull-down samples shared some common subunits in the SDS-PAGE. Closer inspection showed that the PSI and PSII pull-down samples include several same proteins, e.g. PsaA, PsaB, PsaD, PsaF, PsaL, CP47 and PsbO. Some of these subunits were also found in the Cyt $b_6f$ pull-down samples. In addition, the putative NDH-1 subunits NdhK and NdhL were also present in both PSI and Cyt $b_6f$ pull-down samples. Overall, the presence of subunits from unlabelled

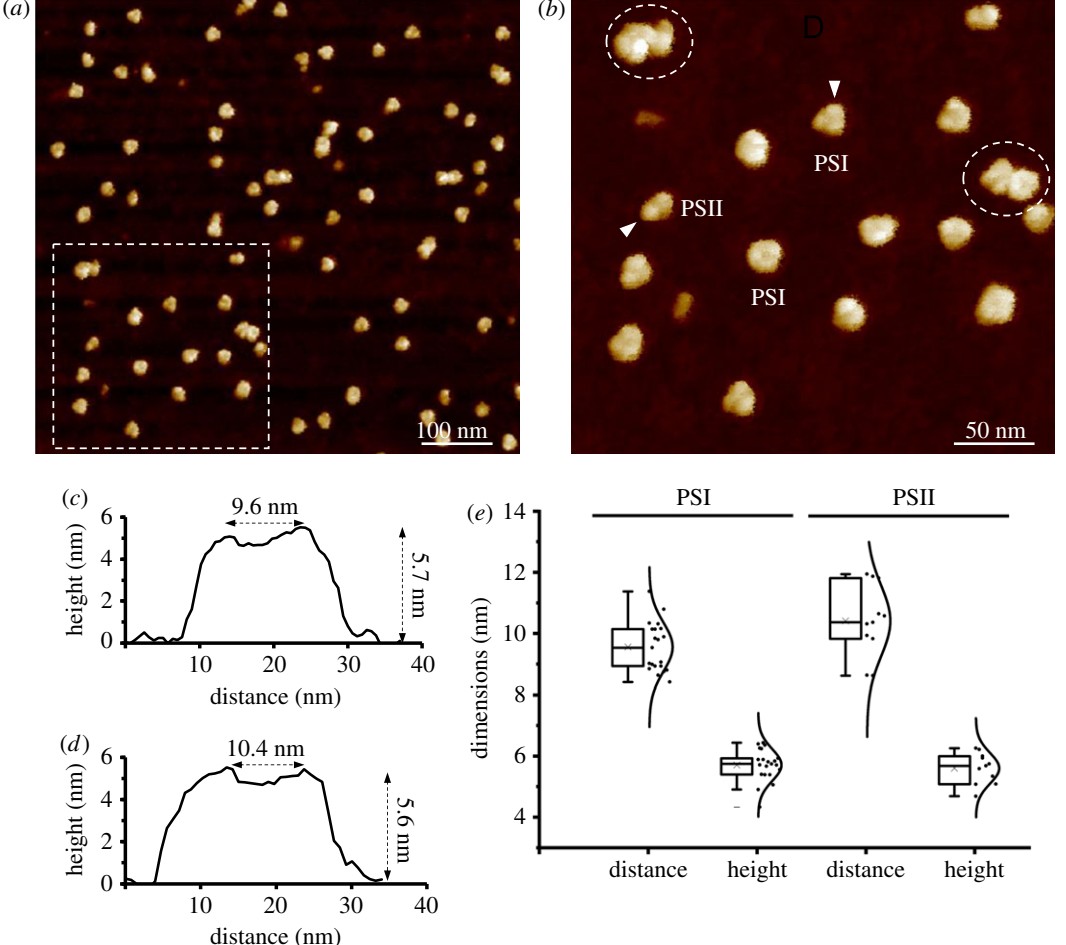

**Figure 2.** AFM imaging of isolated thylakoid membrane complexes from Syn7942. (*a*) Overview AFM image of isolated thylakoid membrane complexes after 1% digitonin membrane solubilization. (*b*) Closer-up AFM image of isolated thylakoid membrane complexes including PSI, PSII and supercomplexes (circles) from (*a*, boxed). (*c*) Cross-section profile analysis of the PSI trimer shown in *b* along the arrow direction. (*d*) Cross-section profile analysis of the PSII dimer shown in *b* along the arrow direction. (*e*) Measurement of the vertex distances and heights of assigned PSI and PSII complexes in AFM. The PSI trimer has a height of $5.7 \pm 0.5$ nm and a distance of $9.6 \pm 0.8$ nm between its two vertexes ($n = 21$). The PSII dimer exhibits a height of $5.6 \pm 0.5$ nm and a vertex distance of $10.4 \pm 1.16$ nm between two protrusions ($n = 11$).

complexes in the GFP-labelled sample indicated the specific association between photosynthetic complexes and potentially the formation of supercomplexes.

## 3.2. Mass spectrometry reveals the physical associations of photosynthetic complexes

To determine the presence and abundance of protein subunits in the pull-down samples, we applied label-free mass spectrometry and relative quantification. A total of 180 protein subunits were detected (electronic supplementary material, file S1), of which 38 subunits belong to the four photosynthetic complexes. The GFP abundance varied among the GFP-labelled PSI, PSII, Cyt $b_6f$ and ATPase pull-down samples (electronic supplementary material, file S2). The GFP abundance was the highest in the PSI pull-down sample, followed by those in the PSII, Cyt $b_6f$ and then the ATPase pull-down samples. The variation is consistent with the variation of the content of photosynthetic complexes in Syn7942 as reported previously [3]. For comparison, the abundance of individual protein subunits was normalized against GFP content in each sample. As a negative control, most of the subunits exhibited a notably low abundance in the WT sample, implicating the specificity of GFP immunoprecipitation (figure 4, grey dots).

As the photosynthetic complexes PSI, PSII, Cyt $b_6f$ and ATPase are multi-subunit complexes, we calculated the sum of the abundance of all the protein subunits in one specific complex to represent the quantity of this complex [42]. Assuming if 100% PSII form stable supercomplexes with PSI with a ratio of $1:1$, the abundance of PSI in the GFP−PSII sample is identical to the abundance of PSI in the

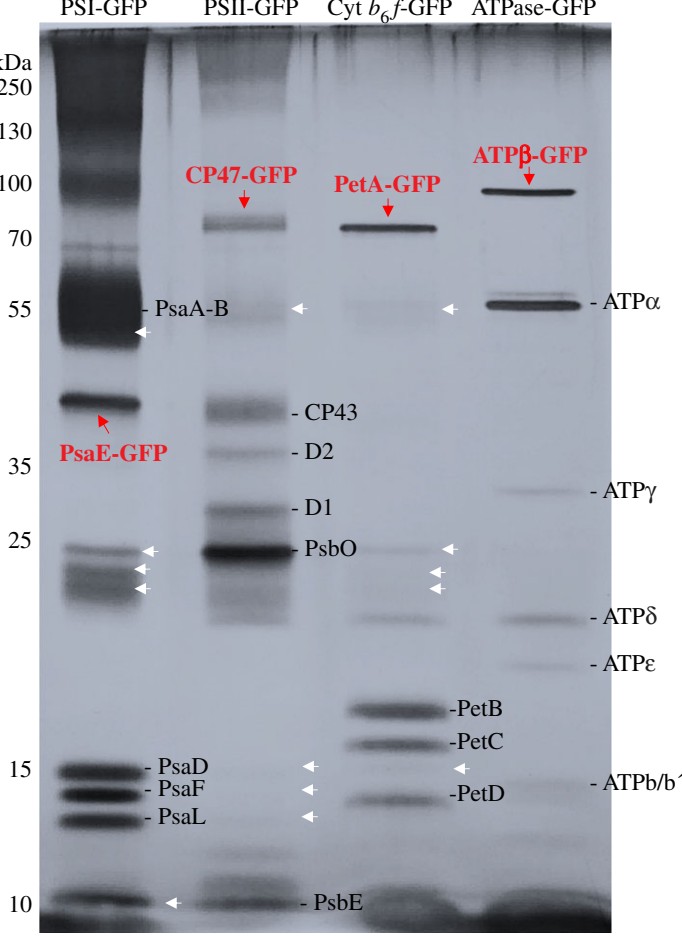

**Figure 3.** SDS-PAGE analysis of the pull-down samples, revealing the subunit composition of GFP-tagged protein complexes and interacting proteins. Black lines indicate the designated subunits in individual complexes. Red arrows indicate the corresponding subunits tagged with GFP in individual complexes. White arrows indicate the potential interacting proteins to the corresponding GFP-tagged complexes.

GFP-PSI sample. However, not all PSII form supercomplexes with PSI in reality. By comparing the abundance of specific complex A between the samples in which GFP was tagged to A and the pull-down sample in which GFP was tagged with the interacting complex B, we could estimate the ratio of the protein complex B that closely associates with A.

Although it is likely that some complex−complex associations could disassemble during purification due to their weak inter-complex interactions, our result showed that PSI and PSII have a stronger preference to form supercomplexes (figure 5). Approximately 1.7% of PSI are associated with PSII (figure 5$a$), whereas 9.9% of PSII strongly interact with PSI (figure 5$b$), presumably owing to a high PSI/PSII ratio of 4.5 in Syn7942 [3]. AFM imaging on thylakoid membranes from Syn7942 confirmed that a large amount of PSI associates tightly with each other to form PSI-enriched membrane regions [20]. Also, 4.1% Cyt $b_6f$ tend to bind with PSI that accounts for 0.3% of the total PSI, whereas 2.0% Cyt $b_6f$ are associated with PSII complexes that are 1.98% of the total PSII (figure 5$c$). These results support the supercomplex formation between PSI and PSII [18] and between photosystems and Cyt $b_6f$ [21]. All PSI, PSII and Cyt $b_6f$ show a low preference to interact with ATPases, while ATPases exhibit relatively similar tendencies to associate with the other three complexes (figure 5$d$).

## 3.3. The binding sites of photosynthetic complexes

By determining the relative abundance of peptides that associate with the GFP-labelled protein subunits, we also evaluated the inter-complex interactions and the binding sites of complex assemblies. The relative abundance of each protein subunit was determined by dividing the abundance of this subunit in a sample with that of the same protein subunit in the GFP-labelled pull-down samples (see

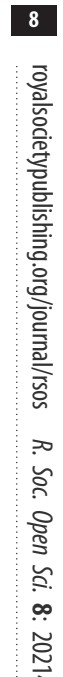

**Figure 4.** Proteins identified by mass spectrometry and comparisons with the WT control. Proteins in the pull-down samples from PSI–GFP (*a*), PSII–GFP (*b*), Cyt $b_6f$–GFP (*c*) and ATPase–GFP (*d*) are compared with the WT control plotted according to their statistical $\log_{10}$ *p*-value (*y*-axis) and their relative abundance ratio ($\log_2$ fold change). Compared to the WT, proteins that are more abundant in the pull-down samples are coloured in green and that are significantly less in the pull-down samples are shown in red. Proteins with more than 20-fold change are labelled with their gene names. Grey dots below the lines (corresponding to $p = 0.05$) are proteins with no significant difference.

Material and methods). Subsequently, the relative abundance of individual subunits was compared within the same protein complex (figure 6).

Within the PSI subunits, the peripheral subunit PsaK showed a greater relative abundance than other subunits in the PSII, Cyt $b_6f$ and ATPase–GFP pull-down samples, suggesting that PsaK may be the primary binding site of PSI to other photosynthetic complexes (figure 6*a*). Consistently, PsaK has been found to be involved in PSI–IsiA binding in Syn7942 under ion-stressed conditions [6] and Lhca3/ Lhca2 binding in *Arabidopsis* [43]. Within the PSII subunits, PsbU, PsbV and PsbN showed a higher abundance in other samples than in the PSII–GFP sample (figure 6*b*). PsbU and PsbV are close to each other and are both extrinsic components of PSII on the lumen side of the thylakoid membrane. They may not have direct interactions with PSI. But the associations with other photosynthetic complexes presumably have impacts on the binding of PsbU and PsbV to PSII, which is important for the regulation of PSII stability and energy transfer [44–46]. PsbN has yet been identified in the existing PSII structure [10,47]; in tobacco, PsbN is not a constituent subunit of PSII but is involved in repair from photoinhibition and assembly of the PSII reaction centre [48]. The peripheral transmembrane subunit PsbY of PSII has also a higher ratio compared to many other PSII subunits. The function of PsbY has not been thoroughly studied, but it was proposed that PsbY is required for

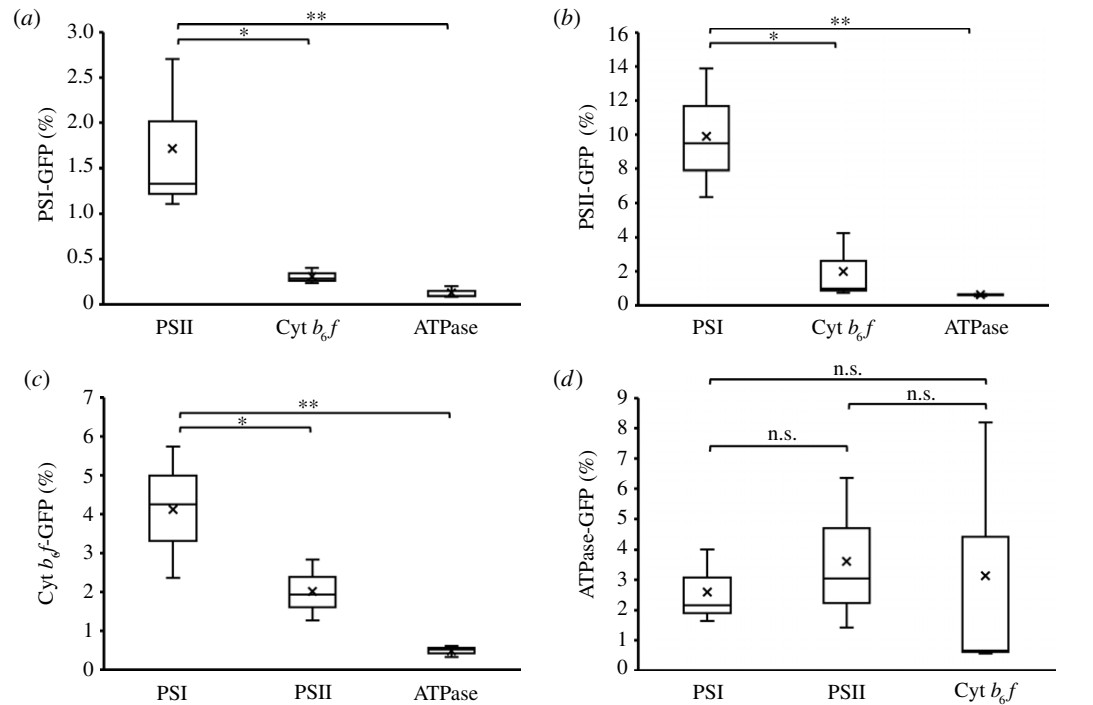

**Figure 5.** Ratios of complexes that tend to associate with other complexes. After normalizing protein content by the GFP content in individual pull-down samples, the percentages of the photosynthetic complexes that associate with other complexes were calculated by dividing the sum of the subunit abundance of a specific photosynthetic complex detected in the pull-down samples for another GFP-labelled complex with the sum of the abundance of the same photosynthetic complex from its corresponding GFP-labelled pull-down sample. For example, the ratio of PSII that associates with PSI is determined by the abundance of PSII in the PSI–GFP pull-down sample divided by the PSII content in the PSII–GFP pull-down sample. The results showed that PSI and PSII possess a higher preference to form supercomplexes than with others (1.72% PSI are associated with PSII, $n = 3$; 9.9% PSII are associated with PSI, $n = 3$) (a,b). Cyt $b_6f$ has a higher preference to bind with PSI (4.1%, $n = 3$) than PSII (2.0%, $n = 3$) (c). All PSI, PSII and Cyt $b_6f$ have a low preference to interact with ATPases, while ATPases exhibit relatively similar tendencies to associate with the other three complexes (d). Box plots display the median (line), the average (cross), the interquartile range (box), and the maximum and minimum (whiskers). Statistical analysis was performed using Student's $t$-test. $^*0.01 < p < 0.05$; $^{**}0.001 < p < 0.01$; n.s., not significant.

the prevention of photodamage to PSII under high light [49], the condition that favours the formation of PSI–PSII supercomplexes [21].

The PetM subunit of Cyt $b_6f$ appears to have a strong preference to interact with other complexes (figure 6c). Indeed, deletion of *petM* in *Synechocystis* sp. PCC 6803 could result in the reduced content of PSI and phycobilisomes, and PetM is involved in the regulatory processes of electron transfer pathways [50]. In ATPase, the subunit b′ possesses a significant difference in the relative abundance (figure 6d). The long α-helices of the peripheral stalk subunits b′ and b expose to the outside of ATPase and clamp the integral membrane subunit a in its position next to the c-ring rotor, connecting $F_1$ to $F_o$ [51]. Moreover, in the rotation of ATPase during ATP production, subunit b′ remains stable, thus making it favourable to bind with other complexes.

Based on the results of pull-down assays and mass spectrometry, we proposed the models of photosynthetic complex associations (figure 7). The peripheral subunit PsaK of PSI, PetM of Cyt $b_6f$ and the subunit b′ of ATPase may sit at the interfaces between distinct photosynthetic complexes to facilitate specific associations of photosynthetic complexes in thylakoid membranes (figure 7a–d). AFM imaging on the native photosynthetic membranes has revealed the structures, lateral distribution and physical associations of photosynthetic complexes [3,20]. Similar associations of different photosynthetic complexes as the proposed models have been observed in AFM (figure 7e–h), although the structural variability could be the intricate properties of supercomplex assemblies in thylakoid membranes [20]. The structural variability might be more significant in PSII associations, resulting in unidentified specific binding sites in the PSII complex.

There is increasing experimental evidence as to the formation of electron transport supercomplexes in cyanobacterial and chloroplast thylakoids [17–19,21,22,41,52,53] and mitochondria [54]. The associations

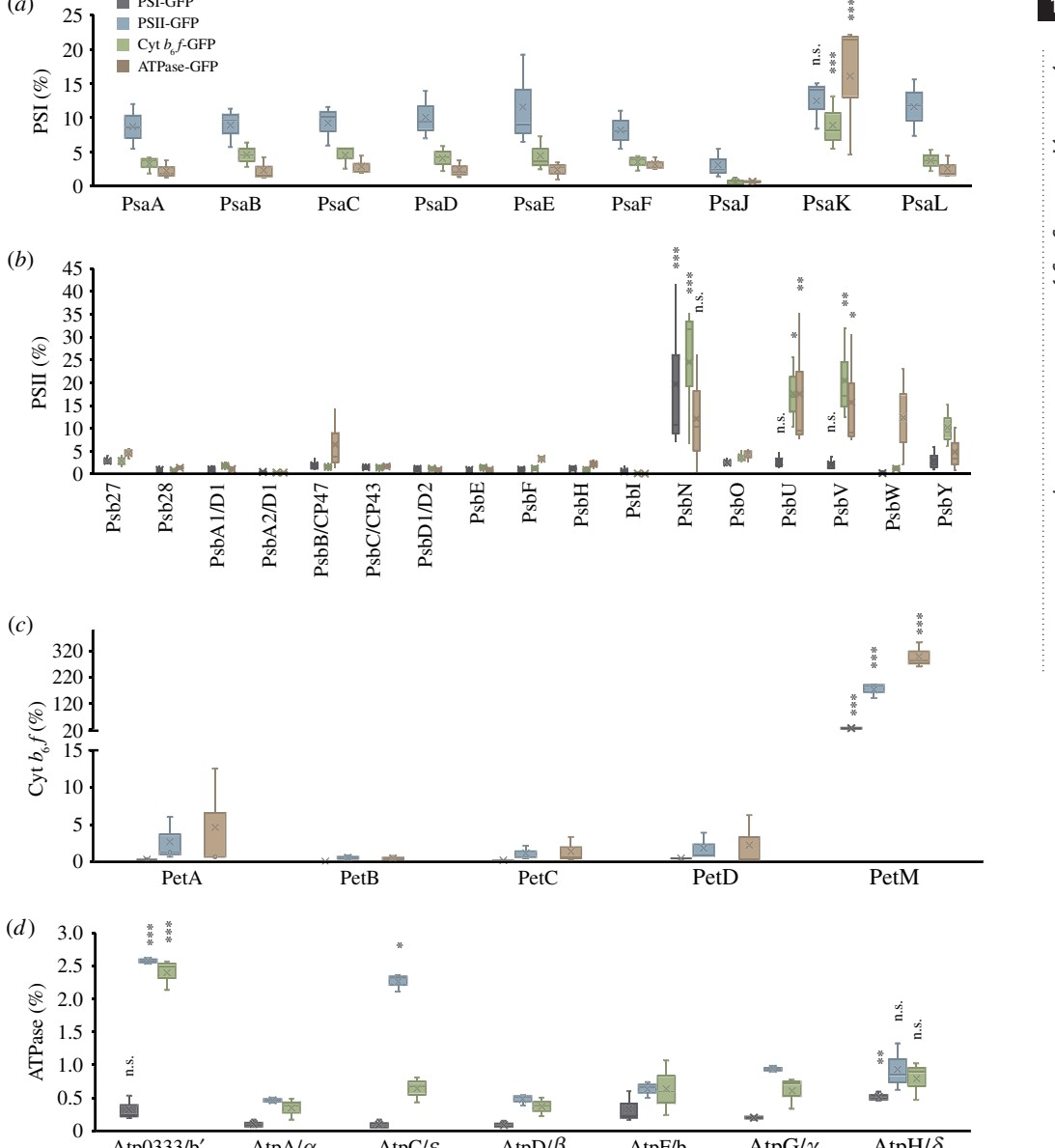

**Figure 6.** Ratios of individual subunits in the corresponding photosynthetic complexes, which are involved in inter-complex associations. The analysis was conducted for the PSI–GFP (*a*), PSII–GFP (*b*), Cyt $b_6f$–GFP (*c*) and ATPase–GFP (*d*) pull-down samples. The relative abundance of each subunit was calculated by determining the ratio of the abundance score of the same subunit in a specific sample to that in the sample in which GFP was labelled to the photosynthetic complex that this subunit belongs to. Statistical analysis was performed using Student's *t*-test to compare the ratio of each subunit with the ratios of other subunits within the same photosynthetic complexes in the same pull-down sample. $^*0.01 < p < 0.05$; $^{**}0.001 < p < 0.01$; $^{***}p < 0.001$; n.s., not significant. Box plots display the median (line), the average (cross), the interquartile range (box) and the maximum and minimum (whiskers).

between different photosynthetic complexes in thylakoid membranes appear weak, transient and dynamic [20], given the highly dynamic and regulatable thylakoid membrane environment in cyanobacteria [2]. This makes it challenging to identify and isolate functionally active supercomplexes for the following investigation. Further improvement is required to reinforce the inter-complex associations, such as using cross-linking reagents [18].

## 3.4. Association between PSI and NDH-1

In addition to the supercomplexes formed by photosynthetic complexes, PSI–NDH-1 supercomplexes have also been characterized in cyanobacteria and plants to facilitate cyclic electron transport

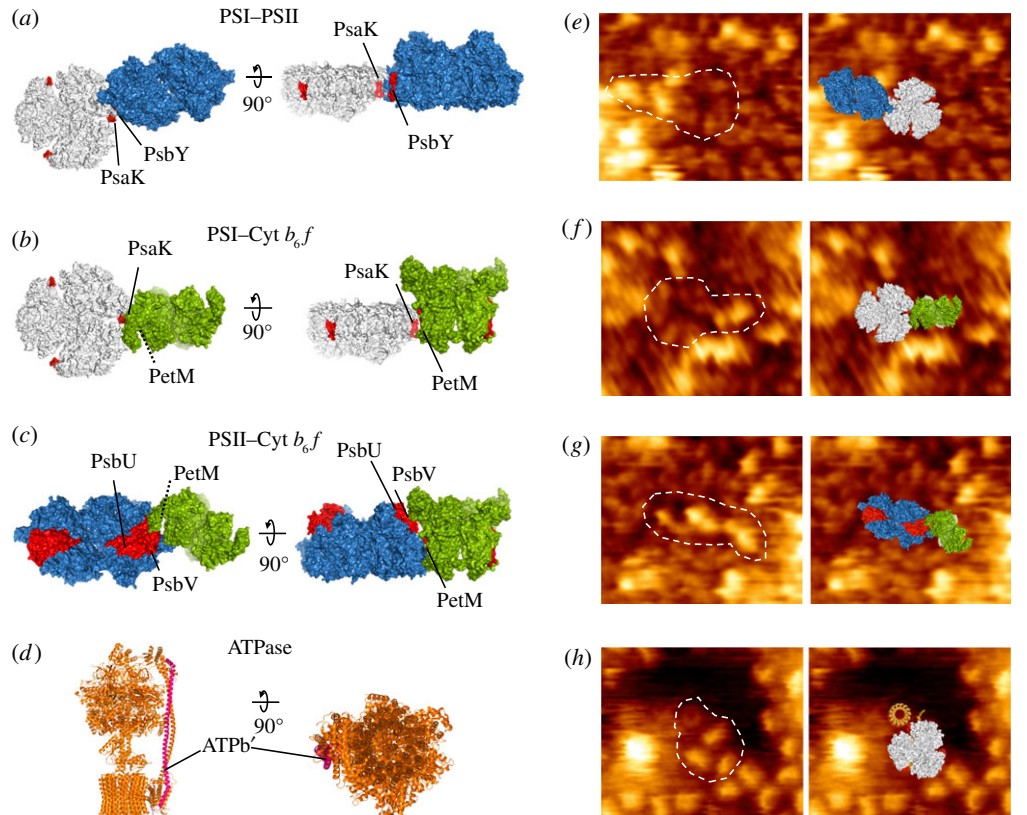

**Figure 7.** Hypothetical models of photosynthetic supercomplex associations. Protein structures were obtained from PDB (PSI: 1JB0; PSII: 3WU2, Cyt $b_6f$: 4H13, ATPase: 6FKF). (a) The hypothetical model of PSI–PSII supercomplex. Possible interacting subunits are PsaK of PSI and PsbY of PSII (red). (b) The hypothetical model of PSI–Cyt $b_6f$ supercomplex. They interact at PsaK of PSI and PetM of Cyt $b_6f$ (red). (c) The hypothetical model of PSII–Cyt $b_6f$ supercomplex. They interact at PsbU, PsbV of PSII and PetM of Cyt $b_6f$ (red). (d) The location of subunit b' (pink) in ATPase. (e,f) AFM visualization of PSI–PSII from the lumenal surface (e), PSI–Cyt $b_6f$ from the lumenal surface (f), PSII–Cyt $b_6f$ from the lumenal surface (g), and PSI–ATPase from the cytoplasmic surface (h) associations, corresponding to (a–d).

[19,20,22]. Our results exhibited that NdhA, NdhM and NdhN have a significantly higher abundance in the PSI–GFP pull-down sample than in other samples (figure 8a). The structure of NDH-1 revealed that NdhA forms a part of a heel to support a peripheral arm of the complex (Q-module) that comprises photosynthesis-specific subunits NdhM, NdhN, NdhO and NdhS [12–16]. The peripheral arm is proven to be the site that interacts with ferredoxin (Fd) to plastoquinone (PQ) [12,14]. The binding of PSI to this part of NDH-1 appears to be functionally preferable. AFM confirmed that such PSI–NDH-1 associations do exist in native thylakoid membranes from Syn7942 (figure 8b,c), resembling the plant PSI–NDH-1 supercomplex observed by cryo-EM [22]. Consistently, higher light can trigger the redistribution of NDH-1 in Syn7942 cells from patches to more even distribution, co-localized with PSI, along the thylakoid membranes [23]. This suggested the closer proximity of NDH-1 complexes to PSI, which probably correlates with a switch from linear to cyclic photosynthetic electron transport.

## 3.5. Additional proteins interacting with the major photosynthetic complexes

Other than the photosynthetic complexes and NDH-1 complexes described above, we have also detected additional proteins from the GFP pull-down samples (electronic supplementary material, table S1). The iron stress-induced chlorophyll-binding protein IsiA and PSI form a supercomplex to increase the absorption cross-section of PSI [5,6,20,55–57]. In addition, the bicarbonate transporter subunit CmpA, the chromosome partition protein Smc, the secretion protein HlyD, peptidoglycan glycosyltransferase were shown to bind to PSI. Additional proteins associating with PSII include three FtsH proteins that are involved in the repair of photodamaged PSII [58,59], band 7 proteins Sea0026 and Sea0027 as two prohibitin homologues that may prevent degradation of newly synthesized D1 [60], the PSI assembly protein Ycf4, and the bicarbonate transporter SbtA. Cyt $b_6f$ may interact with DNA topoisomerase

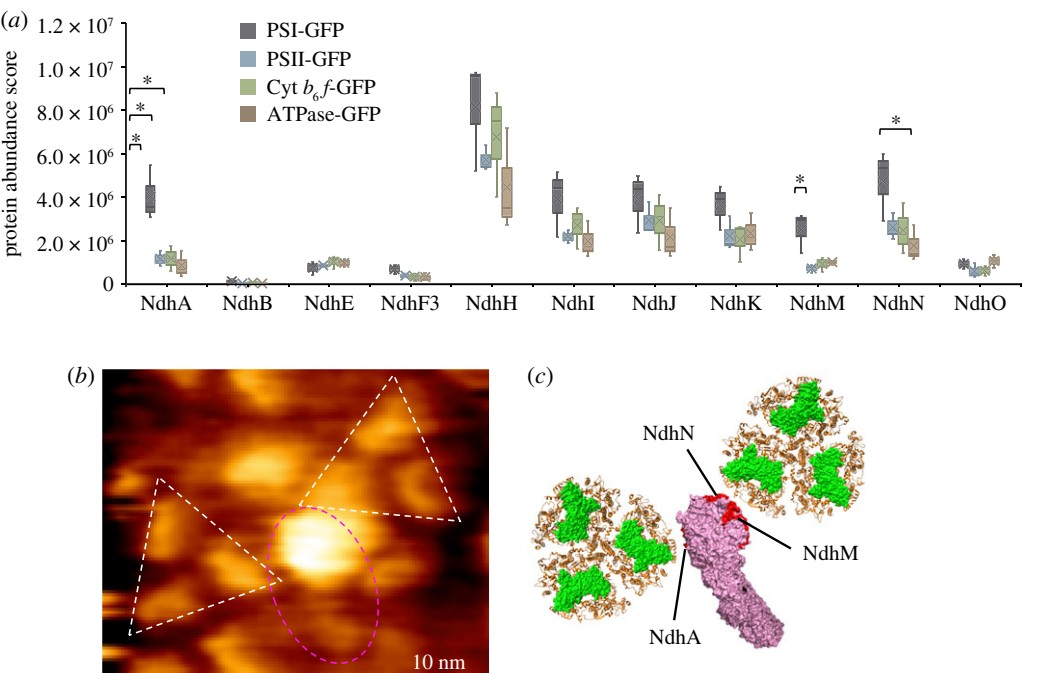

**Figure 8.** NDH-1 subunits detected in four pull-down groups and the hypothetical model of PSI–NDH-1 association. (*a*) NDH-1 subunits detected in each GFP-labelled pull-down samples. Box plots display the median (line), the average (cross), the interquartile range (box) and the maximum and minimum (whiskers). Statistical analysis was performed using Student's *t*-test. *$0.01 < p < 0.05$. (*b*) AFM image of the PSI–NDH-1 associations in native thylakoid membranes from Syn7942. (*c*) The hypothetical model of PSI–NDH-1 associations based on the AFM image shown in (*b*). The potential interacting sites NdhA, NdhN and NdhM of NDH-1 are shown in red. The protein structures were obtained from PDB (PSI: 1JB0, NDH-I: 6NBQ).

TopA and geranylgeranyl reductase, which catalyse the synthesis of phytyl pyrophosphate required for the production of chlorophylls, phylloquinone and tocopherols [61]. ATPase has a close relationship with the chaperon protein DnaK3 [62], 30S ribosomal protein, 50S ribosomal protein, a transcriptional regulator AbrB, etc.

# 4. Conclusion

In summary, we describe a method that integrates data derived from GFP pull-down assays, mass spectrometry and AFM for biochemical and structural characterization of photosynthetic complex assemblies from thylakoid membranes of Syn7942. We characterized in detail the specific inter-complex associations and potential binding domains of different photosynthetic complexes, and proposed the structural models of photosynthetic complex associations. Moreover, other binding proteins that associate with the major photosynthetic complexes were also identified, indicating the highly connecting networks in the thylakoid membrane. Our study delivers insight into the physical interplay of photosynthetic complexes and partners in cyanobacterial thylakoid membranes, providing the structural basis for efficient energy transfer. Advanced knowledge of the molecular basis underlying the organization and interactions of protein complexes in the photosynthetic machinery will inform strategies for the rational design and engineering of artificial photosynthetic membranes and light-driven charge separation systems, with the intent of efficiently capturing and stabilizing solar energy to underpin energy production.

Data accessibility. The datasets supporting this article have been uploaded as part of the electronic supplementary material.

Authors' contributions. Z.Z. performed sample preparation, pull-down assays and analysis of the results; L.-S.Z. carried out AFM imaging, and Z.Z. and L.-S.Z. performed AFM data analysis; L.-N.L. conceived and supervised the work; Z.Z. and L.-N.L. wrote the manuscript.

Competing interests. At the time of writing, Prof. Lu-Ning Liu is a board member of Royal Society Open Science, but had no involvement in the review or assessment of the paper.

Funding. This work was supported by the Royal Society University Research Fellowship (grant no. URF\R\180030 to L.-N.L.) and Biotechnology and Biological Sciences Research Council Grant (grant nos. BB/R003890/1, BB/M024202/1, BB/V009729/1 to L.-N.L.).

Acknowledgements. We acknowledge Dr Catarina Franco and Prof. Robert Beynon at the Centre for Proteome Research, University of Liverpool for support of mass spectrometry analysis. We acknowledge the Liverpool Centre for Cell Imaging for provision of AFM equipment and technical assistance.

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
