## [Peer Review File · Royal Society Open Science]

Review History

RSOS-202142.R0 (Original submission)

Review form: Reviewer 1

Is the manuscript scientifically sound in its present form?

Yes

Are the interpretations and conclusions justified by the results?

Yes

Is the language acceptable?

Yes

Do you have any ethical concerns with this paper?

No

Have you any concerns about statistical analyses in this paper?

No

Recommendation?

Accept with minor revision (please list in comments)

Comments to the Author(s)

Manuscript RSOS-202142 describes a method to investigate supercomplex formation in the thylakoid membrane of cyanobacterium *Synechococcus elongatus* PCC 7942 by using a combination of pull-down assays with eGFP-tagged proteins, mass spectrometry and atomic force microscopy. Zhang et al focus in their work on the five different complexes: Photosystem I (PSI) and II (PSII), cytochrome b6f, ATP synthase and NDH-1 which were tagged with eGFP in a previous study by the PI's lab. Using their methodology, the authors identify proteins that could potentially be involved in supporting the formation of supercomplexes, such as PsaK which might allow the binding of PSI to other photosynthetic complexes. The data is clearly presented, and I find the methodology to be of great use for future studies in S7942 and other cyanobacteria.

Major comments:

Introduction:

The author's statement that all photosystems are located in intracellular thylakoid membranes (line 34) needs to be corrected. The cyanobacterium *Gloeobacter* has no thylakoid membranes but PSI and PSII in the plasma membrane (see e.g. Rexroth et al., 2011, *Plant Cell* (DOI: <https://doi.org/10.1105/tpc.111.085779>)). I also find the statement that thylakoids are parallel to the cytoplasmic membrane (line 36) highly debatable. While it is certainly the case for *Synechococcus elongatus* PCC 7942 many other cyanobacteria show less well organised membrane structures, such as *Anabaena* sp. PCC 7120 (see e.g. Magnuson and Cardona, 2016, *BBA* (DOI: <https://doi.org/10.1016/j.bbabi.2015.10.016>)). Furthermore, the choice of the selected references for the structures in lines 42-43 is unclear and should be revised.

Materials and Methods:

My initial understanding was that the eGFP-tagged mutants were generated in this study as the method is mentioned in line 70. However, in the results section (lines 170-171) it becomes clear that the mutants were generated in an earlier study by the PI's lab. This should be clearly stated in the material and methods section. The current version is somewhat confusing.

Several methods lack detail.

Line: 76: Centrifugation details need to be provided

Line 78: How was the cell disruption performed? Please specify the device and other details such as the number of cycles and their duration.

Line 81: The supplier of digitonin should be provided. It is furthermore unclear, how successful the solubilisation was. Did the authors also test different concentrations of detergent? It is known from other studies that high detergent concentrations can lead to the loss of oligomerisation e.g. in PSI (see e.g. Li et al., 2019, *Nature Plants* (DOI: <https://doi.org/10.1038/s41477-019-0566-x>))

Line 81: "Vortex at minimum speed" needs to be specified.

Line 89,92: Please specify whether the "membrane suspension buffer" is part of the kit or was prepared by the authors (composition?).

Line 90: How was the SDS-PAGE performed? Please indicate the type of gel and the buffer system.

Lines 91-92: Since detailed volumes are given for the mass spectrometry approach it is important to clarify the volume that was used to elute (or wash?) the beads.

Line 130: It is unclear which other unlabelled strains the authors used for comparison. From the previous section I assumed that only the WT was used.

Line 151: Why do the authors use a different buffer for the imaging approach by AFM?

Line 160: Please indicate the composition of the buffer that was used for the gradient centrifugation and how many fractions were collected. Showing photos of the gradients in a supplementary figure would be helpful.

Results and Discussion:

While the AFM images shown in Figure 2 are truly beautiful it is unclear to me how the distance between the vertexes and height can be used to distinguish PSI and PSII as there seems to be no significant difference. How can it be excluded that the particles are just in another orientation when imaged?

Related to the description of complexes, it appears to me that the PSI trimers in Figure 7 are all slightly different. To me it seems to be more obvious that the lower spot (monomer) in the white circle belongs to a trimer outside of the white circle. The authors should explain in more detail how the oligomers (trimers) are selected.

The molecular weight of the eGFP-tagged proteins shown in Figure 3 and described in line 187 do not match. The ATP β -GFP (85 kDa) runs much higher than CP47-GFP (88.5 kDa). There are also several bands that are not highlighted by any arrow in Figure 3 and not discussed. It would be crucial to clarify the weight discrepancy and include the visible bands in the discussion.

Since the data presented for the PsbY interaction with PSI seems to be not statistically supported it should be deleted or phrased more carefully (line 241).

Other minor comments:

Line 11: The keyword "Mass spectroscopy" should be changed to "Mass spectrometry"

Line 26: Should read: ...physiological adaptation of the photosynthetic apparatus.

Line 35: Consider changing "granum" to "grana"

Line 55: The term "communicating complexes" should be clarified.

Line 96: ...heated to 80°C

Line 100: ... of TFA at 37°C

Line 105: What does FA stand for?

Line 122: Please add space between PCC and 7942.

Line 164: A reference for ImageJ should be provided.

References: Some authors are missing from reference number 14.

Review form: Reviewer 2

Is the manuscript scientifically sound in its present form?

No

Are the interpretations and conclusions justified by the results?

No

Is the language acceptable?

Yes

Do you have any ethical concerns with this paper?

No

Have you any concerns about statistical analyses in this paper?

No

Recommendation?

Major revision is needed (please make suggestions in comments)

Comments to the Author(s)

The manuscript of Zhang et al. entitled "Characterizing the supercomplex association of photosynthetic complexes in cyanobacteria" reports about a combined mass spectrometry and atomic force microscopy approach to identify inter-complex associations between the major photosynthetic membrane protein complexes in cyanobacteria. Individually tagged protein complexes were immunoprecipitated by GFP pull-down after membrane solubilization with digitonin and analyzed by mass spectrometry for the presence of subunits from other complexes. Complementary, solubilized membranes were analyzed by atomic force microscopy for the presence of supercomplexes. How membrane protein supercomplexes are formed in cyanobacteria is an important question and the approach is interesting. However, I have doubts about the interpretation of the results.

My biggest concern is the specificity of the presented interactions between the complexes. Are these real supercomplexes or artefacts due to the detergent solubilization and 'stickiness' of subunits? This fundamental problem is not solved by the experimental approach in this study. Particularly, because only partial complexes or few subunits have been identified in the pull-down experiments and not entire complexes with all subunits in stoichiometric amounts, which I would expect for a supercomplex.

Therefore, I suggest the following points to improve the study:

- a) The authors should include all identified proteins in the analysis (Figs. 5/6) and not only those of photosynthetic complexes to provide an unbiased view of the results.
- b) The AFM analysis should be done after the pull-down, maybe in a complementary tagging approach (different tags on different complexes), to get a clearer picture of the potential supercomplex formation.
- c) The authors may think about including crosslinking and mass spectrometry to identify more specific interactions and to get also structural information about the potential supercomplexes.

Decision letter (RSOS-202142.R0)

Dear Professor Liu

The Editors assigned to your paper RSOS-202142 "Characterizing the supercomplex association of photosynthetic complexes in cyanobacteria" have now received comments from reviewers and would like you to revise the paper in accordance with the reviewer comments and any comments from the Editors. Please note this decision does not guarantee eventual acceptance.

Please submit your revised manuscript and required files (see below) no later than 21 days from today's (ie 01-Mar-2021) date. Note: the ScholarOne system will 'lock' if submission of the revision is attempted 21 or more days after the deadline. If you do not think you will be able to meet this deadline please contact the editorial office immediately.

on behalf of Professor Diwakar Shukla (Associate Editor) and Catrin Pritchard (Subject Editor)
openscience@royalsociety.org

Reviewer comments to Author:

Reviewer: 1

Comments to the Author(s)

Manuscript RSOS-202142 describes a method to investigate supercomplex formation in the thylakoid membrane of cyanobacterium *Synechococcus elongatus* PCC 7942 by using a combination of pull-down assays with eGFP-tagged proteins, mass spectrometry and atomic force microscopy. Zhang et al focus in their work on the five different complexes: Photosystem I (PSI) and II (PSII), cytochrome b6f, ATP synthase and NDH-1 which were tagged with eGFP in a previous study by the PI's lab. Using their methodology, the authors identify proteins that could potentially be involved in supporting the formation of supercomplexes, such as PsaK which might allow the binding of PSI to other photosynthetic complexes. The data is clearly presented, and I find the methodology to be of great use for future studies in S7942 and other cyanobacteria.

Major comments:

Introduction:

The author's statement that all photosystems are located in intracellular thylakoid membranes (line 34) needs to be corrected. The cyanobacterium *Gloeobacter* has no thylakoid membranes but PSI and PSII in the plasma membrane (see e.g. Rexroth et al., 2011, *Plant Cell* (DOI: <https://doi.org/10.1105/tpc.111.085779>)). I also find the statement that thylakoids are parallel to the cytoplasmic membrane (line 36) highly debatable. While it is certainly the case for *Synechococcus elongatus* PCC 7942 many other cyanobacteria show less well organised membrane structures, such as *Anabaena* sp. PCC 7120 (see e.g. Magnuson and Cardona, 2016, *BBA* (DOI: <https://doi.org/10.1016/j.bbabi.2015.10.016>)). Furthermore, the choice of the selected references for the structures in lines 42-43 is unclear and should be revised.

Materials and Methods:

My initial understanding was that the eGFP-tagged mutants were generated in this study as the method is mentioned in line 70. However, in the results section (lines 170-171) it becomes clear

that the mutants were generated in an earlier study by the PI's lab. This should be clearly stated in the material and methods section. The current version is somewhat confusing.

Several methods lack detail.

Line 76: Centrifugation details need to be provided

Line 78: How was the cell disruption performed? Please specify the device and other details such as the number of cycles and their duration.

Line 81: The supplier of digitonin should be provided. It is furthermore unclear, how successful the solubilisation was. Did the authors also test different concentrations of detergent? It is known from other studies that high detergent concentrations can lead to the loss of oligomerisation e.g. in PSI (see e.g. Li et al., 2019, Nature Plants (DOI: <https://doi.org/10.1038/s41477-019-0566-x>))

Line 81: "Vortex at minimum speed" needs to be specified.

Line 89,92: Please specify whether the "membrane suspension buffer" is part of the kit or was prepared by the authors (composition?).

Line 90: How was the SDS-PAGE performed? Please indicate the type of gel and the buffer system.

Lines 91-92: Since detailed volumes are given for the mass spectrometry approach it is important to clarify the volume that was used to elute (or wash?) the beads.

Line 130: It is unclear which other unlabelled strains the authors used for comparison. From the previous section I assumed that only the WT was used.

Line 151: Why do the authors use a different buffer for the imaging approach by AFM?

Line 160: Please indicate the composition of the buffer that was used for the gradient centrifugation and how many fractions were collected. Showing photos of the gradients in a supplementary figure would be helpful.

Results and Discussion:

While the AFM images shown in Figure 2 are truly beautiful it is unclear to me how the distance between the vertexes and height can be used to distinguish PSI and PSII as there seems to be no significant difference. How can it be excluded that the particles are just in another orientation when imaged?

Related to the description of complexes, it appears to me that the PSI trimers in Figure 7 are all slightly different. To me it seems to be more obvious that the lower spot (monomer) in the white circle belongs to a trimer outside of the white circle. The authors should explain in more detail how the oligomers (trimers) are selected.

The molecular weight of the eGFP-tagged proteins shown in Figure 3 and described in line 187 do not match. The ATP β -GFP (85 kDa) runs much higher than CP47-GFP (88.5 kDa). There are also several bands that are not highlighted by any arrow in Figure 3 and not discussed. It would be crucial to clarify the weight discrepancy and include the visible bands in the discussion.

Since the data presented for the PsbY interaction with PSI seems to be not statistically supported it should be deleted or phrased more carefully (line 241).

Other minor comments:

Line 11: The keyword "Mass spectroscopy" should be changed to "Mass spectrometry"

Line 26: Should read: ...physiological adaptation of the photosynthetic apparatus.

Line 35: Consider changing "granum" to "grana"

Line 55: The term "communicating complexes" should be clarified.

Line 96: ...heated to 80°C

Line 100: ... of TFA at 37°C

Line 105: What does FA stand for?

Line 122: Please add space between PCC and 7942.

Line 164: A reference for ImageJ should be provided.

References: Some authors are missing from reference number 14.

Reviewer: 2

Comments to the Author(s)

The manuscript of Zhang et al. entitled "Characterizing the supercomplex association of photosynthetic complexes in cyanobacteria" reports about a combined mass spectrometry and atomic force microscopy approach to identify inter-complex associations between the major photosynthetic membrane protein complexes in cyanobacteria. Individually tagged protein complexes were immunoprecipitated by GFP pull-down after membrane solubilization with digitonin and analyzed by mass spectrometry for the presence of subunits from other complexes. Complementary, solubilized membranes were analyzed by atomic force microscopy for the presence of supercomplexes. How membrane protein supercomplexes are formed in cyanobacteria is an important question and the approach is interesting. However, I have doubts about the interpretation of the results.

My biggest concern is the specificity of the presented interactions between the complexes. Are these real supercomplexes or artefacts due to the detergent solubilization and 'stickiness' of subunits? This fundamental problem is not solved by the experimental approach in this study. Particularly, because only partial complexes or few subunits have been identified in the pull-down experiments and not entire complexes with all subunits in stoichiometric amounts, which I would expect for a supercomplex.

Therefore, I suggest the following points to improve the study:

- a) The authors should include all identified proteins in the analysis (Figs. 5/6) and not only those of photosynthetic complexes to provide an unbiased view of the results.
- b) The AFM analysis should be done after the pull-down, maybe in a complementary tagging approach (different tags on different complexes), to get a clearer picture of the potential supercomplex formation.
- c) The authors may think about including crosslinking and mass spectrometry to identify more specific interactions and to get also structural information about the potential supercomplexes.

===PREPARING YOUR MANUSCRIPT===

===PREPARING YOUR REVISION IN SCHOLARONE===

Author's Response to Decision Letter for (RSOS-202142.R0)

See Appendix A.

RSOS-202142.R1 (Revision)

Review form: Reviewer 1

Is the manuscript scientifically sound in its present form?

Yes

Are the interpretations and conclusions justified by the results?

Yes

Is the language acceptable?

Yes

Do you have any ethical concerns with this paper?

No

Have you any concerns about statistical analyses in this paper?

No

Recommendation?

Accept as is

Comments to the Author(s)

I would like to thank the authors for addressing all comments in such detail.

Decision letter (RSOS-202142.R1)

Dear Professor Liu,

It is a pleasure to accept your manuscript entitled "Characterizing the supercomplex association of photosynthetic complexes in cyanobacteria" in its current form for publication in Royal Society Open Science. The comments of the reviewer(s) who reviewed your manuscript are included at the foot of this letter.

on behalf of Professor Diwakar Shukla (Associate Editor) and Catrin Pritchard (Subject Editor)
openscience@royalsociety.org

Reviewer comments to Author:
Reviewer: 1
Comments to the Author(s)

I would like to thank the authors for addressing all comments in such detail.

Appendix A

Responses to Reviewers

We sincerely appreciate the reviewers' extremely positive comments on our work and efforts to review our manuscript.

Reviewer: 1

Manuscript RSOS-202142 describes a method to investigate supercomplex formation in the thylakoid membrane of cyanobacterium *Synechococcus elongatus* PCC 7942 by using a combination of pull-down assays with eGFP-tagged proteins, mass spectrometry and atomic force microscopy. Zhang et al focus in their work on the five different complexes: Photosystem I (PSI) and II (PSII), cytochrome *b₆f*, ATP synthase and NDH-1 which were tagged with eGFP in a previous study by the PI's lab. Using their methodology, the authors identify proteins that could potentially be involved in supporting the formation of supercomplexes, such as PsaK which might allow the binding of PSI to other photosynthetic complexes. The data is clearly presented, and I find the methodology to be of great use for future studies in S7942 and other cyanobacteria.

Major comments:

Introduction:

The author's statement that all photosystems are located in intracellular thylakoid membranes (line 34) needs to be corrected. The cyanobacterium *Gloeobacter* has no thylakoid membranes but PSI and PSII in the plasma membrane (see e.g. Rexroth et al., 2011, *Plant Cell* (DOI: <https://doi.org/10.1105/tpc.111.085779>)).

Response: We have revised the statement as "In most cyanobacteria, light-dependent reactions ...".

I also find the statement that thylakoids are parallel to the cytoplasmic membrane (line 36) highly debatable. While it is certainly the case for *Synechococcus elongatus* PCC 7942 many other cyanobacteria show less well organised membrane structures, such as *Anabaena* sp. PCC 7120 (see e.g. Magnuson and Cardona, 2016, *BBA* (DOI: <https://doi.org/10.1016/j.bbabi.2015.10.016>)).

Response: We have revised the statement as "instead, they generally form stacks of membrane layers that sit between the cytoplasmic membrane and central cytoplasm [1,2]".

Furthermore, the choice of the selected references for the structures in lines 42-43 is unclear and should be revised.

Response: Thanks for pointing this out. We intended to list the reported structures of photosynthetic complexes from cyanobacteria. We have revised this sentence as follows to clarify the cyanobacterial origins of the selected membrane complexes.

"the atomic structures of major photosynthetic membrane complexes from cyanobacteria have been resolved, including Photosystem I (PSI) [4-6], Photosystem II (PSII) [7-10], Cytochrome *b₆f* (Cyt *b₆f*) [11], and photosynthetic Complex I – NAD(P)H dehydrogenase (NDH-1) [14-18]."

Materials and Methods:

My initial understanding was that the eGFP-tagged mutants were generated in this study as the method is mentioned in line 70. However, in the results section (lines 170-171) it becomes clear that the mutants were generated in an earlier study by the PI's lab. This should be clearly stated in the material and methods section. The current version is somewhat confusing.

Response: Thanks for pointing this out. We have added "The GFP-labeled Syn7942 strains have been generated in our previous study [3]" in Materials and Methods to clarify this.

Several methods lack detail.

Line: 76: Centrifugation details need to be provided.

Response: We have added the centrifugation speed and duration in the revised manuscript.

Line 78: How was the cell disruption performed? Please specify the device and other details such as the number of cycles and their duration.

Response: We have added cell disruption details in the revised manuscript.

Line 81: The supplier of digitonin should be provided. It is furthermore unclear, how successful the solubilisation was. Did the authors also test different concentrations of detergent? It is known from other studies that high detergent concentrations can lead to the loss of oligomerisation e.g. in PSI (see e.g. Li et al., 2019, Nature Plants (DOI: <https://doi.org/10.1038/s41477-019-0566-x>))

Response: We have revised it as “1% digitonin (Sigma-Aldrich)”. We agree with the reviewer that digitonin at different concentrations could result in different levels of membrane solubilization. Therefore, we have tested the digitonin concentration ranging from 0 to 2%. The resulting supernatant samples were examined by electron microscopy to ensure both membrane solubilization and membrane complex intactness (Figure 2). Eventually, 1% digitonin was chosen as the optimal condition for thylakoid membrane solubilization. We have added the detailed descriptions about the process in Materials and Methods.

Line 81: “Vortex at minimum speed” needs to be specified.

Response: We have provided detailed descriptions as “with vortex at 2,700 rpm for 5 times 1 min on and 1 min off and then 10 times 30 s on and 30 s off.”

Line 89,92: Please specify whether the “membrane suspension buffer” is part of the kit or was prepared by the authors (composition?).

Response: We are sorry for the typo. It should be “membrane resuspension buffer” as we mentioned earlier (see 3.2), and its composition is 10 mM Tris pH 6.8, 200 mM NaCl, 1 mM EDTA. We have corrected it in the revised manuscript.

Line 90: How was the SDS-PAGE performed? Please indicate the type of gel and the buffer system.

Response: We have provided detailed descriptions of SDS-PAGE in Materials and Methods.

Lines 91-92: Since detailed volumes are given for the mass spectrometry approach it is important to clarify the volume that was used to elute (or wash?) the beads.

Response: We have provided a detailed description in Materials and Methods as follows:

“For SDS-PAGE analysis, proteins were eluted with 50 μ L elution buffer following the instructions provided by the manufacturer. Then 10 μ L samples were loaded onto a 12% SDS gel with 4 \times sample buffer (1.57% Tris-HCl pH 6.8, 4% SDS (w/v), 20% glycerol, 0.1% bromophenol blue (w/v), 1.5% dithiothreitol (w/v)). For mass spectrometry, bound proteins and beads were collected by taking the column out of the magnetic field and were washed with 50 μ L the membrane resuspension buffer. 10 μ L was used for SDS-PAGE and the rest 40 μ L was used for proteomic analysis”

Line 130: It is unclear which other unlabelled strains the authors used for comparison. From the previous section I assumed that only the WT was used.

Response: We have revised this sentence to clarify this issue as follows: “Ratios of complexes involved in supercomplex formation were calculated by comparing the abundance of specific protein subunits between the GFP-labeled strain and other strains in which the specific subunits were not labeled with GFP using GFP normalized data”. In addition, an example has been explained in the same section.

Line 151: Why do the authors use a different buffer for the imaging approach by AFM?

The electrostatically balanced interactions between AFM tip and native biological macromolecules are crucial for reducing mechanical damage to biological structures and achieving high-resolution

AFM topographs in aqueous solution. Based on the previous studies (Biophysical J 1999, 76: 1101-1111) and our experience in AFM imaging on photosynthetic apparatus (Nature Plants 2020, 6: 869-882; Mol Plant 2017, 10: 1434-1448; PNAS 2011, 108:9455-9459; J Struct Biol 2011, 173: 138-145; J Mol Biol 2009, 393: 27-35), the adsorption buffer (10 mM Tris-HCl pH 7.5, 150 mM KCl, 25 mM MgCl₂) and imaging buffer (10 mM Tris-HCl pH 7.5, 150 mM KCl) that we used in this AFM study has been applied as the optimal buffers to ensure high-resolution AFM imaging of photosynthetic complexes at the near physical condition. We have added the descriptions in Materials and Methods.

Line 160: Please indicate the composition of the buffer that was used for the gradient centrifugation and how many fractions were collected. Showing photos of the gradients in a supplementary figure would be helpful.

Response: The buffer used in sucrose gradient for thylakoid isolation was based on 50 mM MES-NaOH pH 6.5, 5 mM CaCl₂, and 10 mM MgCl₂, with the addition of sucrose. The isolated thylakoid membranes at 1.3–2.0M sucrose fraction were collected for AFM imaging. The step sucrose gradient centrifugation of thylakoid membranes has been illustrated in our previous studies (Nature Plants 2020, 6: 869-882; Mol Plant 2017, 10: 1434-1448). We have added the corresponding details and references in the revised Materials and Methods.

Results and Discussion:

While the AFM images shown in Figure 2 are truly beautiful it is unclear to me how the distance between the vertexes and height can be used to distinguish PSI and PSII as there seems to be no significant difference. How can it be excluded that the particles are just in another orientation when imaged?

Response: As explained in the figure legend, the height profiles and distance between the vertexes of PSI and PSII particles were taken following the white arrows shown in panel (b). The distance between two monomers within the PSI trimers is shorter than that between the two monomers within the PSII dimers. We agree with the reviewer that it is possible that PSI and PSII have both orientations exposing to AFM probes. However, in our experiments we found that the relatively flat surfaces of PSI (the lumen side) and PSII (the cytoplasmic side) are prone to attach to the mica surface, exposing the more protruded surfaces to the AFM probe. This thus facilitates the profile and distance analysis of individual photosynthetic complexes.

Related to the description of complexes, it appears to me that the PSI trimers in Figure 7 are all slightly different. To me it seems to be more obvious that the lower spot (monomer) in the white circle belongs to a trimer outside of the white circle. The authors should explain in more detail how the oligomers (trimers) are selected.

Response: We would like to address that the “lower spots” represent the structural features of the PSI trimer luminal surface, as we have reported in our recent study (see Fig. 6 in Nature Plants 2020, 6: 869-882, which has been also copied below). The PSI trimeric structures were assigned based on the high-resolution AFM images of typical PSI protrusions. As the reviewer suggested, we have revised the figure legend to provide more detailed descriptions of the AFM images.

Fig. 6 (shown in Nature Plants 2020, 6: 869-882). AFM images reveal PSII and Cyt b_6f in thylakoid membranes from ML-adapted Syn7942. a, High-resolution AFM image of the luminal surface of thylakoid membranes, showing the densely packed photosynthetic membrane proteins. The area represented by the white box is shown in b. b, Zoomed-in view of the area highlighted in a. PSI trimers are highlighted with white dashed circles based on their unique topography as shown in d. Putative PSII and Cyt b_6f complexes are highlighted by a green and pink oval, respectively, based on the distance between their two monomers from the luminal membrane surface. White arrows indicate the positions of height profiles. c, Height profiles corresponding to PSII and Cyt b_6f in b. The lateral distance between peaks of PSII is 8.9 ± 0.9 nm, $n = 15$, and the height of protrusions from the membrane surface is 3.5 nm. The lateral distance between peaks of Cyt b_6f is 6.2 ± 0.7 nm, $n = 15$, and the height of protrusions from the membrane surface is 3.0 nm. d, Atomic structure (left), simulated AFM images based on PDB (middle) and AFM topograph (right) of PSI, PSII and Cyt b_6f from the luminal surface (PDB, PSI: 1JB0; PSII: 3WU2; Cyt b_6f : 2E74). The lateral size of PSI crystal structure and distances of protrusions in the PSII and Cyt b_6f crystal structures are shown. e, Model of the arrangement of PSI, PSII and Cyt b_6f within the thylakoid membrane, constructed with simulated AFM images of PSI (green), PSII (blue) and Cyt b_6f (purple).

The molecular weight of the eGFP-tagged proteins shown in Figure 3 and described in line 187 do not match. The ATP β -GFP (85 kDa) runs much higher than CP47-GFP (88.5 kDa). There are also several bands that are not highlighted by any arrow in Figure 3 and not discussed. It would be crucial to clarify the weight discrepancy and include the visible bands in the discussion.

Response: SDS-PAGE can be used to estimate the molecular weights of protein subunits based on protein migration. However, it is not an accurate method for determining the absolute molecular weight. We could not exclude the potential effects in SDS-PAGE analysis caused by protein post-translational modification and the higher content of basic amino acids in peptides. To avoid the confusion, we have revised the sentence in text by removing the estimated molecular weight values.

For this same reason, in Figure 3 we only highlighted those proteins that can be confidently designated. It could be overambitious if we designate those bands that may be a mixture of peptides with similar MW or having the low abundance. To identify all the visible bands, we will cut the bands from the gel and conduct gel digestion followed by mass spectrometry in future studies.

Since the data presented for the PsbY interaction with PSI seems to be not statistically supported it should be deleted or phrased more carefully (line 241).

Response: We agree with the review and have revised the sentence carefully.

Other minor comments:

Line 11: The keyword “Mass spectroscopy” should be changed to “Mass spectrometry”

Response: This has been corrected.

Line 26: Should read: ...physiological adaptation of the photosynthetic apparatus.

Response: Corrected.

Line 35: Consider changing “granum” to “grana”

Response: Corrected.

Line 55: The term “communicating complexes” should be clarified.

Response: We have change it to “... complexes interact and cooperate with”.

Line 96: ...heated to 80°C

Response: Corrected.

Line 100: ... of TFA at 37°C

Response: We have changed it to “trifluoroacetic acid (TFA) at 37°C”.

Line 105: What does FA stand for?

Response: We have changed it to “formic acid (FA)”.

Line 122: Please add space between PCC and 7942.

Response: Corrected.

Line 164: A reference for ImageJ should be provided.

Response: We have added the following reference for ImageJ in the revised manuscript: “Schneider, C. A.; Rasband, W. S. & Eliceiri, K. W. (2012) NIH Image to ImageJ: 25 years of image analysis. Nature methods 9(7): 671-675, PMID 22930834.”

References: Some authors are missing from reference number 14.

Response: Thanks for pointing this out. The Endnote format provided for Royal Society Open Science has been automatically set to show the first few authors. We have edited to show all authors in references.

Reviewer: 2

The manuscript of Zhang et al. entitled “Characterizing the supercomplex association of photosynthetic complexes in cyanobacteria” reports about a combined mass spectrometry and atomic force microscopy approach to identify inter-complex associations between the major photosynthetic membrane protein complexes in cyanobacteria. Individually tagged protein complexes were immunoprecipitated by GFP pull-down after membrane solubilization with digitonin and analyzed by mass spectrometry for the presence of subunits from other complexes.

Complementary, solubilized membranes were analyzed by atomic force microscopy for the presence of supercomplexes. How membrane protein supercomplexes are formed in cyanobacteria is an important question and the approach is interesting. However, I have doubts about the interpretation of the results.

My biggest concern is the specificity of the presented interactions between the complexes. Are these real supercomplexes or artefacts due to the detergent solubilization and 'stickiness' of subunits? This fundamental problem is not solved by the experimental approach in this study. Particularly, because only partial complexes or few subunits have been identified in the pull-down experiments and not entire complexes with all subunits in stoichiometric amounts, which I would expect for a supercomplex.

Response: Membrane solubilization by digitonin or DDM have been generally used for isolation and biochemical and structural characterization of various photosynthetic complexes/supercomplexes from different phototrophic organisms, as reported previously (Nature 2016, 534: 69-74; Science 2019, 365: eaax4406; Science 2013, 342: 1104-1107). In addition, tagging of photosynthetic complexes has been applied to investigate inter-complex interactions of PSI and PSII (Mol Plant 2017, 10: 62-72; Photosynthetica. 2018, 56: 300-305). Although we cannot exclude any effects caused by detergent treatment and isolation in determining protein interactions, our developed approaches could provide a means for identifying proteins involved in the formation of supercomplexes and the weak, transient interactions between photosynthetic complexes. This is based on the high specificity of tags and pull down as well as the high sensitivity of mass spectrometry. As Reviewer 1 acknowledged, "the methodology to be of great use for future studies in S7942 and other cyanobacteria".

In our experiments, the detected photosynthetic protein subunits in different pull-down samples varied in abundance and we have involved proper unlabeled samples as controls to confirm the sensitivity and accuracy of our approach. We further used AFM imaging on isolated complexes and natural thylakoid membranes to validate the association of these supercomplexes.

We confess some protein peptides are difficult to digest by enzymes and detect in MS. Due to the different degrees of stability, it is possible that some protein complexes or intermediates could disassemble during membrane solubilization and pull-down assays. This could likely lead to partial complexes or some subunits detected in the pull-down experiments, as we discussed in the manuscript. Further technical improvement is required to establish a more sensitive and reliable method for studying complex-complex interactions, involving using cross-linking and genetic modification.

Therefore, I suggest the following points to improve the study:

- a) The authors should include all identified proteins in the analysis (Figs. 5/6) and not only those of photosynthetic complexes to provide an unbiased view of the results.

Response: We would like to emphasize that Figure 4 has shown all the MS-identified proteins, and the detailed data analysis and MS datasets have been provided in Supplementary Table 1 and Supplementary files 1 and 2. The present study focuses on the assembly of photosynthetic complexes. Figures 5 and 6 were generated using an in-house search and analysis (as described in Materials and Methods) to compare the relative abundance of individual subunits to indicate the potential binding subunits between photosynthetic complexes specifically. Figure 5 shows the different binding affinities between photosynthetic complexes at the complex level, whereas Figure 6 illustrates the individual photosynthetic protein subunits that are involved in inter-complex associations at the single subunit level.

- b) The AFM analysis should be done after the pull-down, maybe in a complementary tagging approach (different tags on different complexes), to get a clearer picture of the potential supercomplex formation.

Response: Due to the technical constraints of the GFP pulldown assay kit, the protein complexes from the antibody-attached beads in the pulldown columns had to be eluted using a denaturation buffer (see Figure 1 in the manuscript, which has been copied below). Thus, we could not apply AFM

imaging on the denatured pull-down samples to visualize supercomplex assemblies. The technique will be further developed to ensure native elution.

Figure 1. Overview of GFP immunoprecipitation methodology. Crude thylakoid membranes from GFP-labeled PSI, PSII, Cyt b_6f , ATPase and WT strains were prepared by centrifugation and were solubilized by 1% digitonin. Proteins that interact with GFP-labeled proteins are immunoprecipitated with GFP pull-down assay and subjected to mass spectrometric analysis. Triplicated experiments were conducted for each strain.

c) The authors may think about including crosslinking and mass spectrometry to identify more specific interactions and to get also structural information about the potential supercomplexes.

Response: We appreciate the reviewer's excellent suggestion and have planned a combination of crosslinking and mass spec analysis in the following studies. Crosslinking does provide strong interactions between supercomplexes. On the other hand, it may also result in unspecific aggregations of protein complexes and therefore artifacts to some degrees.